# Body mass index and physical frailty among older Mexican Americans: Findings from an 18-year follow up

**Megan Rutherford[1], Brian Downer[2], Chih-Ying Li[3], Lin-Na Chou[4], Soham Al Snih [2,5,6]***

**1** School of Medicine, University of Texas Medical Branch, Galveston, TX, United States of America,
**2** Department of Population Health and Health Disparities/School of Public and Population Health, University of Texas Medical Branch, Galveston, TX, United States of America, **3** Department of Occupational Therapy/School of Health Professions, University of Texas Medical Branch, Galveston, TX, United States of America, **4** Department of Biostatistics and Data Science/School of Public and Population Health, University of Texas Medical Branch, Galveston, TX, United States of America, **5** Division of Geriatrics & Palliative Medicine/Department of Internal Medicine, University of Texas Medical Branch, Galveston, TX, United States of America, **6** Sealy Center on Aging, University of Texas Medical Branch, Galveston, TX, United States of America

* soalsnih@utmb.edu

**Data Availability Statement:** Data are available from: https://www.icpsr.umich.edu/icpsrweb/NACDA/series/546.

# Abstract

## Purposes

The relationship between body mass index (BMI) and frailty in older Mexican Americans has not been previously studied. The objective of this study was to examine the relationship between BMI and frailty among non-frail older Mexican Americans at baseline over 18 years of follow up.

## Methods

Longitudinal population-based study of 1,648 non-institutionalized Mexican Americans aged $\geq$ 67 years from the Hispanic Established Population for the Epidemiologic Study of the Elderly (1995/96-2012/13). Frailty phenotype was defined as meeting three or more of the following: unintentional weight loss of >10 pounds, weakness, self-reported exhaustion, low physical activity, and slow walking speed. BMI (kg/m$^2$) was classified as underweight (<18.5), normal weight (18.5-<25), overweight (25-< 30), obesity category I (30-<35), and obesity category II/morbid obesity ($\geq$35). Covariates included socio-demographics, comorbidities, cognitive function, and depressive symptoms. Generalized Estimating Equation models were performed to estimate the odds ratio (OR) and 95% confidence interval (CI) of frailty as a function of BMI category.

## Results

Participants with underweight or obesity category II/ morbid obesity had greater odds of frailty over time compared to those with normal weight (OR = 2.39, 95% CI = 1.29–4.44 and OR = 1.62, 95% CI = 1.07–2.44, respectively) after controlling for all covariates. Participants with BMIs in the overweight or category I obesity were at lower odds of frailty over time.

**Funding:** This work was supported by the National Institute on Minority Health and Health Disparities (R01 MD010355-PI Al Snih S), the National Institute on Aging (R01 AG10939-PI Markides KS, and K01AG058789-PI Downer B), and the National Institute of Child Health and Human Development (K01HD101589-PI Li C). The funders had no role in study design, data collection, and analysis, decision to publish, or preparation of the manuscript.

**Competing interests:** The authors have declared that no competing interests exist.

## Conclusions

Mexican American older adults with BMIs in the underweight or obesity category II/morbid obesity were at higher odds of frailty over time. This indicates that maintaining a healthy weight in this population may prevent future frailty.

## Introduction

Frailty is a geriatric condition characterized by decline in multiple systems leading to an impaired ability to respond to stressors [1]. It predisposes individuals to adverse events including hospitalizations, disability, cognitive decline, falls, and mortality [1–4]. There is no universally recognized definition of frailty, but the frailty phenotype described by Fried et al. is the most commonly used definition to study frailty [1]. Weight loss and low body mass index (BMI) are criteria for some definitions of frailty, including the frailty phenotype [1, 5–7]. However, studies have shown that the prevalence of frailty is higher among those malnourished, but the prevalence of malnourished is less than 10% among older adults with frailty [8–10].

The relationship between BMI and frailty has only recently been explored. Cross-sectional studies have shown that there is a U- or J-shaped relationship between BMI and frailty [6, 11–17]. Ferriolli et al. [13], using the Frailty in Brazilian Elderly (FIBRA-BR) Study, found underweight (BMI $< 18.5$ kg/m$^2$) associated with frailty while obesity (BMI $\geq 30$ kg/m$^2$) was associated with prefrailty. Rietman et al. [14] using the Doetinchem Cohort Study found a high prevalence of frailty among those in the underweight (8.2%) and obesity (5.0%) categories compared with those of normal weight. Findings from the EPIdémiologie De l'OStéoporose, Epidmiology of Osteoporisis (EPIDOS) Study showed a J-shaped relationship between BMI and frailty, where the percentage of underweight and obesity women who were frail were 10.8% and 20.3%, respectively [12].

Longitudinal studies have also found that obesity predicts the development of frailty [18–25]. For example, a study by Stenholm et al. [24] concluded that midlife obesity led to a five-fold increase in the risk of frailty over a 22-year follow up compared to those of normal weight among participants in the Mini-Finland Health Examination Survey. Ho et al. [19] found that older adults in the Taiwan Longitudinal Study on Aging with obesity were twice as likely to develop frailty over 8 years of follow up when compared to a group with high-normal weight. Mezuk et al. [23], examining the trajectories of BMI and incident of frailty among participants in the United States Health and Retirement Study, found that older adults with obesity and weight loss class were almost three times more likely to develop frailty over a 10-year follow-up period when compared to a group that was consistently overweight.

The older Hispanic population is one of the fastest growing ethnic groups in the United States and is projected to rise from 8% in 2018 to 21% in 2060 [26]. This population is characterized by increased risk of diabetes, obesity, and disability, all factors associated with frailty [27, 28]. Research on Mexican Americans has shown an increased risk of frailty in those with older age, female sex, impaired cognitive function, disability, pain, falls, negative affect, and diabetes [3, 4, 28–31]. Previous longitudinal studies examining the relationship between BMI and frailty included mostly non-Hispanic older adults. Therefore, the objective of this study was to examine the relationship between BMI and frailty among non-frail older Mexican Americans at baseline over 18 years of follow up. We hypothesized that Mexican Americans with BMIs in the underweight or obese ranges would have an increased risk of frailty over time compared to Mexican Americans with normal weight.

## Materials and methods

### Data source and study population

The data were from the Hispanic Established Populations for the Epidemiologic Study of the Elderly (EPESE). This is an ongoing longitudinal study of Mexican Americans 65 years and older who reside in Arizona, California, Colorado, New Mexico, and Texas. The original Hispanic EPESE sample consisted of 3,050 participants interviewed at baseline in 1993–1994 and followed-up every 2 or 3 years thereafter. The present study used data collected from wave 2 (1995/96) to wave 8 (2012/13), allowing for approximately 18 years of follow-up data. Information and data for the Hispanic EPESE are available at the National Archive of Computerized Data on Aging [32]. The baseline interview information was used to assess weight loss (a component of the frailty phenotype) as the difference between weight measured in 1993–94 (baseline) and weight measured in 1995–96 (wave 2). Of the 2,438 participants interviewed at wave 2 (hereafter referred as baseline), 860 were non-frail, 835 were pre-frail, and 170 were frail. We excluded the 170 participants who were frail and the 620 with missing information for any covariate, frailty measure, and BMI. The final sample included 1,648 participants aged 67 years and older. Participants excluded from the study were more likely to be older, to have a lower level of education, lower Mini-Mental State Examination (MMSE) score, and lower BMI; they also reported more strokes, heart attacks, cancer, hip fractures, and high depressive symptoms than included participants. At the end of follow up (2013), 348 participants were re-interviewed in person, 259 were lost to follow up or refused to be re-interviewed, and 1,141 were confirmed dead through the National Death Index and report from relatives (Fig 1). The University's Institutional Review Board approved the study protocol, and oral informed consent was obtained from each participant at the time of the interview.

### Measures

#### Predictor variable

**Body mass index assessment (BMI).** BMI was calculated as weight in kilograms divided by height in meters squared ($kg/m^2$). BMI was grouped according to the National Institutes of Health obesity standards: BMI $<18.5$ $kg/m^2$, underweight; 18.5–24.9 $kg/m^2$, normal weight; 25–29.9 $kg/m^2$, overweight; 30–34.9 $kg/m^2$, class I obesity, and BMI $\geq35$ $kg/m^2$, class II/ morbid obesity [33].

#### Outcome variable

**Modified frailty phenotype.** Frailty status was determined by a modified version of the frailty phenotype defined by Fried et al. [1]. The criteria were weight loss, weakness, slowness, low physical activity, and exhaustion [1, 34]. Weight loss: >10 pounds, calculated as the difference between weight measured in 1993/94 and in 1995/96. Weakness: unable to perform the handgrip strength test or in the bottom 20% (adjusted for sex and BMI). Slowness: unable to perform a timed 8-foot walk test or in the bottom 20% (height-sex adjusted). Low physical activity: answered "No" to the question "Can you walk half a mile without help?" (sex-adjusted). Exhaustion: positive responses to questions from the Center for Epidemiologic Studies Depression (CES-D) Scale: "Everything I did was an effort" or "I could not get going." Participants were classified as non-frail if they met none of the criteria, pre-frail if they met 1–2 of the criteria, and frail if they met 3 or more of the criteria. Frailty was assessed at each interview during the 18-year study period.

#### Covariates

Sociodemographic factors assessed were age, sex, years of formal education, and marital status (married vs. not married). Self-reported medical conditions included hypertension, arthritis,

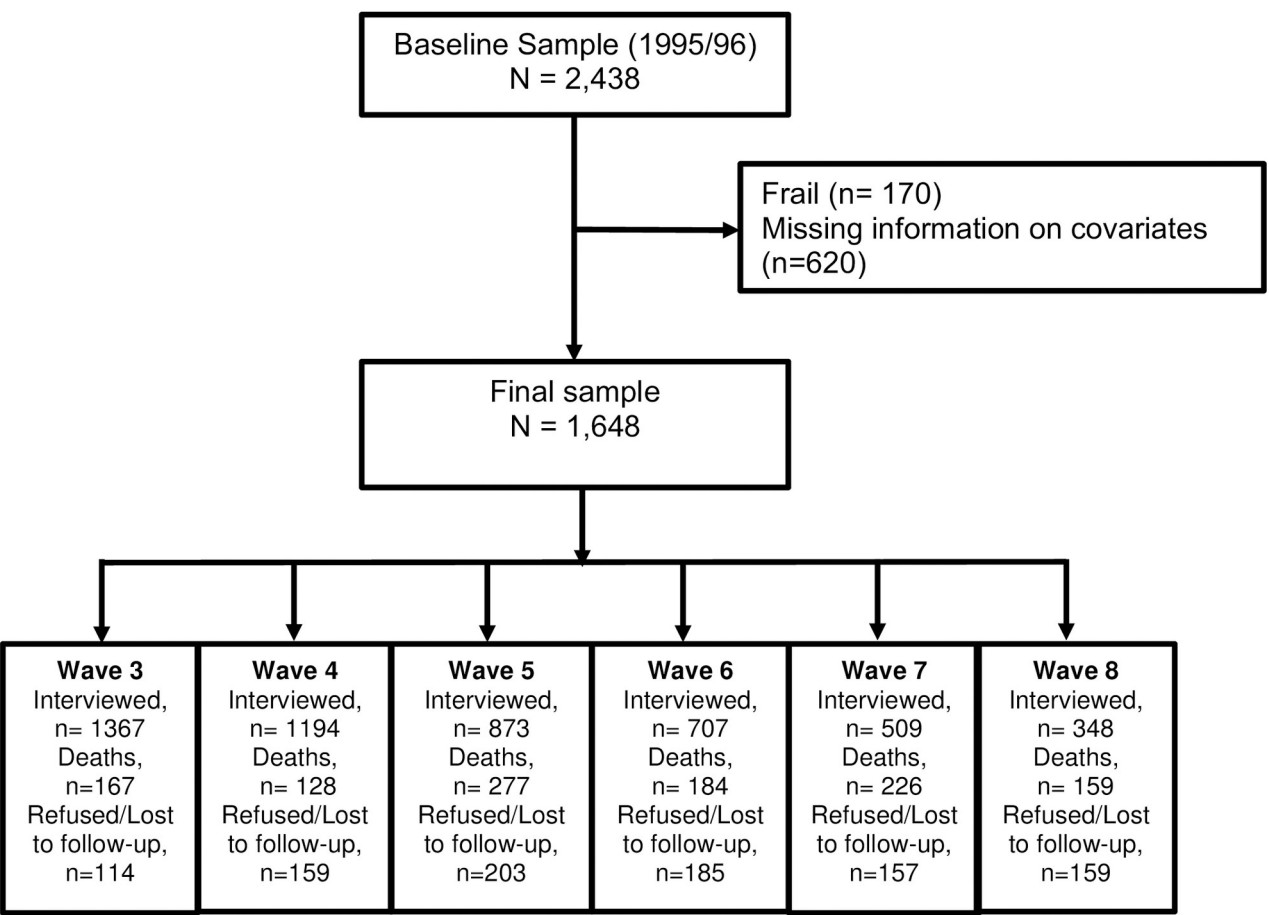

**Fig 1. Sample flow chart.**

diabetes, heart attack, stroke, cancer, and hip fracture. Cognitive function was assessed with the MMSE [35]. Depressive symptoms were measured with the CES-D Scale [36]. A score of ≥16 was used to determine a clinical range for those with depressive symptoms [37].

## Statistical analysis

Chi-square, Fisher's exact, and ANOVA tests were used to describe the sample characteristics by BMI category at baseline. Generalized Estimating Equation using the GENMOD procedure in SAS was used to estimate the odds ratio (OR) and 95% confidence interval (CI) of frailty over 18 years of follow up as a function of BMI category, controlling for socio-demographics, medical conditions, cognitive function, and depressive symptoms. The models used a logit link binomial distribution and autoregressive correlation structure to account for repeated measures of participants. All variables, including BMI categories, were analyzed as time varying (potential to change over time), except for sex and education. Selection bias from missing data is minimizes with the use of GEE models because it allows for the use of all available data from all follow-up interviews while accounting for differences in the follow-up duration. In estimating the working correlation parameters, all non-missing pairs of data taken from the same participants over time are used. Using the GEE procedure, we lose only the observations for which the participant is missing and not all the measurements [38–40]. Participants who died, refused to participate, or were lost to follow up were included until their last follow-up

date (last interview date over the 18-year follow up). Additional analyses excluded those who were prefrail or frail at baseline. All analyses were performed using version 9.4 of SAS (SAS Institute, Inc., Cary, NC, USA).

## Results

Table 1 shows the baseline descriptive characteristics of the overall sample and by BMI category. The mean age was 74.3 [Standard Deviation (SD) = 5.6] years and the mean BMI was 28.0 (SD = 5.1), with participants classified as 1.0% underweight, 27.4% normal weight, 39.8% overweight, 23.2% obesity category I, and 8.6% obesity category II/morbid obesity. Fifty eight percent were female, 54.9% were married, and the mean years of education was 5.0 (SD = 4.0) years. The mean MMSE score was 24.4 points (SD = 4.2). The most common medical conditions were hypertension, arthritis, and diabetes. Nine percent had high depressive symptoms. Those in the obesity category II/morbid obesity were significantly more likely to be younger (72 years), to be female (84.5%), to be unmarried (54.2%), and to report hypertension (64.1%), arthritis (62.0%), diabetes (34.5%), and high depressive symptoms (15.5%) compared to those in the underweight, normal weight, overweight or obesity category I categories. Participants in the overweight and obesity category II/morbid obesity had the highest scores on the MMSE (24.6).

Table 2 presents the results of the generalized estimating equation analysis for frailty over time as a function of BMI category. Participants in the underweight or obesity category II/ morbid obesity categories had greater odds of frailty (OR = 2.40, 95% Confidence Interval (CI)

**Table 1. Sample characteristicsof non-frail older Mexican Americans by BMI categories at baseline (N = 1,648).**

| Variables | Total N (%) | Underweight N (%) | Normal Weight N (%) | Overweight N (%) | Class I Obesity N (%) | Class II/ Morbid Obesity N (%) | p-value |
|---|---|---|---|---|---|---|---|
| **Total** | 1648 (100) | 17 (1.0) | 452 (27.4) | 655 (39.8) | 382 (23.2) | 142 (8.6) | |
| **Age in years, Mean (SD)** | 74.3 (5.6) | 75.6 (5.2) | 75.9 (6.4) | 74.3 (5.3) | 73.1 (5.2) | 72.4 (4.5) | <0.0001 |
| **Sex** | | | | | | | <0.0001 |
| Male | 697 (42.3) | 7 (41.2) | 204 (45.1) | 330 (50.4) | 134 (35.1) | 22 (15.5) | |
| Female | 951 (57.7) | 10 (58.8) | 248 (54.9) | 325 (49.6) | 248 (64.9) | 120 (84.5) | |
| **Education (year)** | 5.0 (4.0) | 4.8 (3.9) | 5.0 (3.8) | 5.0 (4.1) | 5.1 (3.9) | 5.0 (3.8) | 0.9768 |
| **Marital Status** | | | | | | | 0.0480 |
| Married | 905 (54.9) | 9 (52.9) | 237 (52.3) | 384 (58.6) | 210 (55.0) | 65 (45.8) | |
| Unmarried | 743 (45.1) | 8 (47.1) | 215 (47.7) | 271 (41.4) | 172 (45.0) | 77 (54.2) | |
| **BMI (kg/m$^2$), Mean (SD)** | 28.0 (5.1) | 16.8 (1.5) | 22.8 (1.6) | 27.4 (1.4) | 32.0 (1.4) | 38.6 (3.9) | <0.0001 |
| **MMSE Score** | 24.4 (4.2) | 23.6 (4.6) | 23.9 (4.3) | 24.6 (4.1) | 24.5 (4.1) | 24.6 (4.1) | 0.0354 |
| **Medical conditions** | | | | | | | |
| Hypertension | 741 (45.0) | 5 (29.4) | 149 (33.0) | 284 (43.4) | 212 (55.5) | 91 (64.1) | <0.0001 |
| Arthritis | 702 (42.6) | 2 (11.8) | 161 (35.6) | 263 (40.2) | 188 (49.2) | 88 (62.0) | <0.0001 |
| Diabetes | 431 (26.2) | 3 (17.7) | 96 (21.2) | 162 (24.7) | 121 (31.7) | 49 (34.5) | 0.0013 |
| Heart attack | 136 (8.3) | 0 (0.0) | 39 (8.7) | 57 (8.7) | 33 (8.6) | 7 (4.9) | 0.4888 |
| Stroke | 96 (5.8) | 2 (11.8) | 27 (6.0) | 34 (5.2) | 27 (7.1) | 6 (4.2) | 0.4156 |
| Cancer | 99 (6.0) | 0 (0.0) | 33 (7.3) | 35 (5.3) | 20 (5.2) | 11 (7.8) | 0.4462 |
| Hip Fracture | 12 (0.7) | 1 (5.9) | 3 (0.7) | 3 (0.5) | 4 (1.1) | 1 (0.7) | 0.1686 |
| Depressive Symptoms (CES-D) | 151 (9.2) | 0 (0.0) | 31 (6.9) | 54 (8.2) | 44 (11.5) | 22 (15.5) | 0.0065 |

**Note:** BMI = Body Mass Index; SD = standard deviation; MMSE = Mini Mental State Examination; CES-D = Center for Epidemiologic Studies Depression Scale.

Obesity category I = BMI of 30 to < 35 Kg/m$^2$; Obesity category II/morbid obesity = BMI $\geq$ 35 Kg/m$^2$

**Table 2. Generalized estimating equation models for frailty as a function of BMI categories over 18-years of follow up among non-frail older Mexican Americans at baseline (N = 1,648).**

| Variables | Odds Ratio 95% CI | p-value |
|---|---|---|
| **Time** | 1.09 (1.06–1.12) | <0.0001 |
| **BMI categories** | | |
| Underweight | 2.40 (1.29–4.46) | 0.0056 |
| Normal Weight | Reference | Reference |
| Overweight | 0.86 (0.65–1.13) | 0.2706 |
| Obesity Category I | 1.09 (0.80–1.50) | 0.5804 |
| Obesity Category II/Morbid Obesity | 1.62 (1.07–2.44) | 0.0211 |
| **Age (years)** | 1.07 (1.05–1.09) | <0.0001 |
| **Female Sex** | 1.10 (0.84–1.43) | 0.4843 |
| **Years of Education** | 0.99 (0.96–1.02) | 0.5004 |
| **Married** | 1.17 (0.91–1.49) | 0.2134 |
| **MMSE** | 0.92 (0.90–0.94) | <0.0001 |
| **Hypertension** | 0.96 (0.76–1.21) | 0.7460 |
| **Arthritis** | 1.53 (1.22–1.93) | 0.0002 |
| **Diabetes** | 1.20 (0.94–1.55) | 0.1507 |
| **Heart Attack** | 1.60 (1.07–2.39) | 0.0228 |
| **Stroke** | 0.78 (0.48–1.27) | 0.3212 |
| **Cancer** | 1.40 (0.96–2.03) | 0.0829 |
| **Hip Fracture** | 2.56 (1.26–5.20) | 0.0095 |
| **Depressive Symptoms (CES-D $\geq$ 16)** | 4.31 (3.38–5.51) | <0.0001 |

**Note:** BMI = Body Mass Index; MMSE = Mini Mental State Examination; CES-D = Center for Epidemiologic Studies Depression Scale.

Obesity category I = BMI of 30 to < 35 Kg/m$^2$; Obesity category II/morbid obesity = BMI $\geq$35 Kg/m$^2$

= 1.29–4.46 and OR = 1.62, 95% CI = 1.08–2.44, respectively) over time compared to those with normal weight, after controlling for all covariates. Being in overweight or obesity category I did not significantly increase the odds of frailty over time. Older age, arthritis, heart attack, hip fracture, and high depressive symptoms increased the odds of frailty over time. Lower odds of frailty was also observed in those with high MMSE scores. After excluding prefrail or frail participants at baseline (N = 837), those in the underweight or obesity category II/morbid obesity categories had greater odds of frailty over time (OR = 5.09, 95% CI = 1.95–13.32 and OR = 2.80, 95% CI = 1.52–5.14, respectively) compared to those in the normal weight category, after controlling for all covariates. Those in the overweight or obesity category I group were not at increased odds of frailty over time.

Table 3 presents the results of the generalized estimating equation analysis for each frailty criterion over time as a function of BMI category. After controlling for all covariates, underweight participants had greater odds of weight loss (OR = 2.71, 95% CI = 1.71–4.28), overweight or obesity category I participants had greater odds of weakness (OR = 1.44, 95% CI = 1.20–1.71 and OR = 1.62, 95% CI = 1.31–2.01, respectively), those in obesity category I also had greater odds of exhaustion (OR = 1.1, 95% CI = 1.02–1.68), and those in the obesity category II/morbid obesity had greater odds of weakness, slowness, low physical activity, and exhaustion. Those in the overweight, obesity category I or obesity category II/morbid obesity had lower odds of weight loss over time.

Fig 2 shows the participants' frailty status at each wave as a function of BMI category over 18-years of follow up among those who were non-frail at baseline. Frailty increased from

**Table 3. Generalized estimating equation models for each frailty criterion as a function of BMI categories over 18-years of follow up among non-frail older Mexican Americans at baseline (N = 1,648).**

| BMI category | Weight Loss OR (95% CI) | Weakness OR (95% CI) | Slowness OR (95% CI) | Low Physical Activity OR (95% CI) | Exhaustion OR (95% CI) |
|---|---|---|---|---|---|
| Underweight | 2.71 (1.71–4.28) | 1.12 (0.59–2.10) | 0.78 (0.44–1.37) | 1.10 (0.63–1.91) | 1.86 (0.99–3.50) |
| Normal Weight | Reference | Reference | Reference | Reference | Reference |
| Overweight | 0.60 (0.52–0.70) | 1.44 (1.20–1.71) | 0.90 (0.75–1.09) | 1.11 (0.93–1.33) | 1.20 (0.96–1.50) |
| Obesity Category I | 0.54 (0.45–0.64) | 1.62 (1.31–2.01) | 1.03 (0.82–1.29) | 1.22 (0.99–1.51) | 1.31 (1.02–1.68) |
| Obesity Category II /Morbid Obesity | 0.45 (0.34–0.58) | 1.96 (1.47–2.63) | 1.42 (1.06–1.89) | 2.36 (1.80–3.10) | 1.65 (1.17–2.33) |

**Note:** Controlled for time (years) age (years), gender, marital status, years of education, comorbid conditions (hypertension, arthritis, diabetes, heart attack, stroke, cancer, and hip fracture), cognitive function, and depressive symptoms.

BMI = Body Mass Index; OR = odds ratio; CI = confidence interval.

22.2% to 75% for those in the underweight category, 12.2% to 34.6% for those in the normal weight category, 8.7% to 28.8% for those in the overweight category, 6.9% to 37.5% for those in the obesity category I, and 14.9% to 83.3% for those in the obesity category II/morbid obesity.

Fig 3 presents the percent of each frailty criterion by BMI category over time among those who were non-frail at baseline. Weight loss was more prevalent among those who were in the underweight and normal weight categories during the early years of follow up and less prevalent among those who were in the obesity categories. Weakness was consistently more

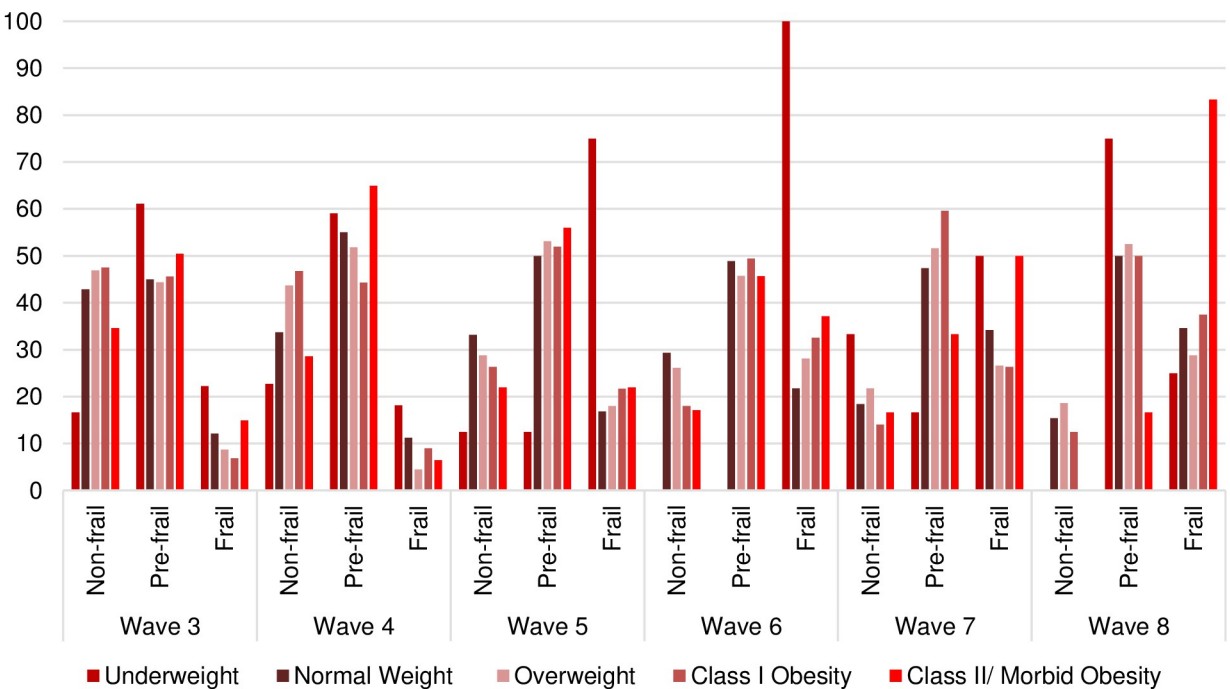

**Note: BMI = Body Mass Index**

**Fig 2. Percent of non-frail, pre-frail, and frailty by BMI categories over time among non-frail older Mexican Americans at baseline (N = 1,648).**

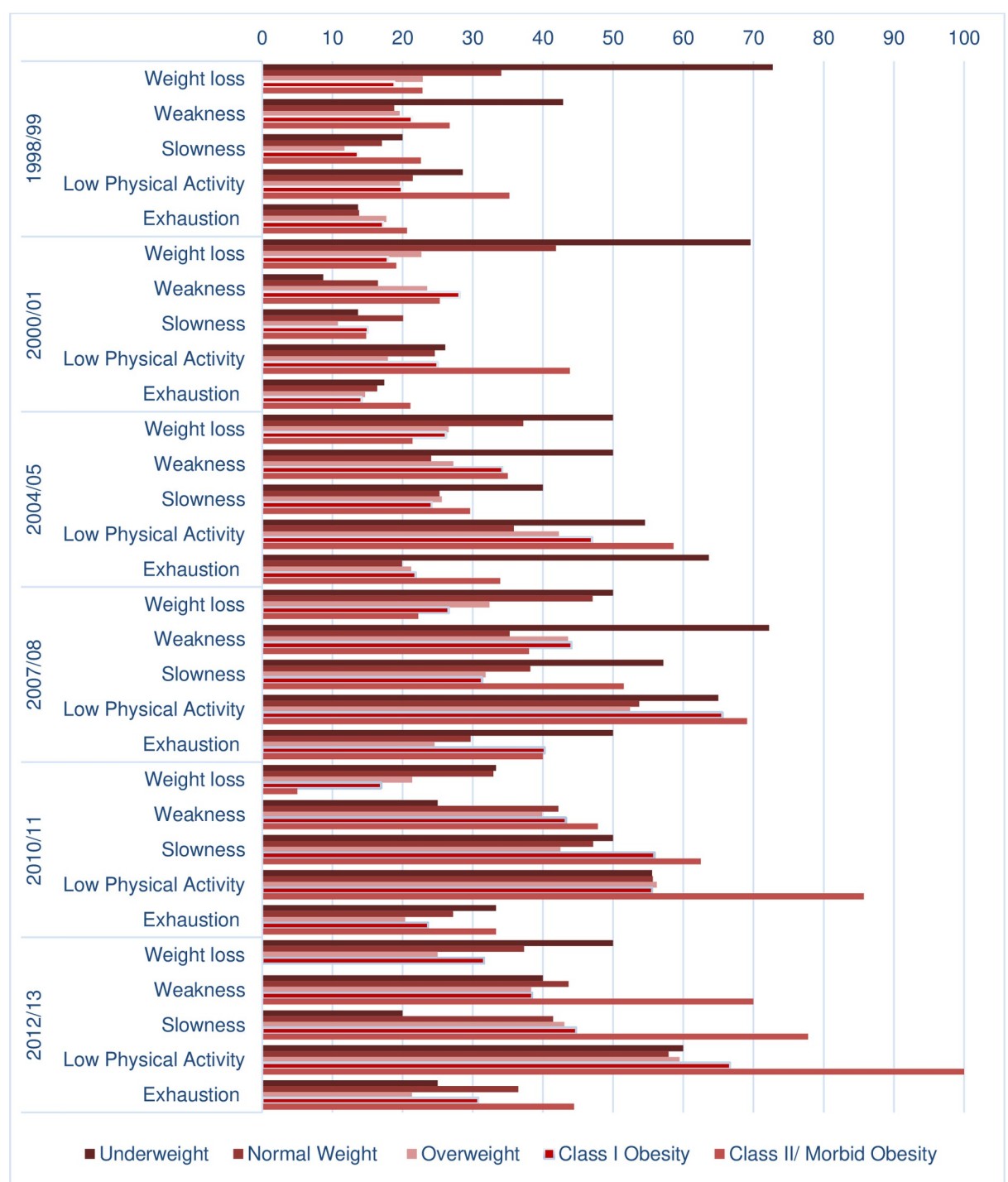

**Note: BMI=Body Mass Index**

**Fig 3. Percent of each frailty criterion by BMI categories over time among non-frail older Mexican Americans at baseline (N = 1,648).**

prevalent among those who were in the underweight category or obesity categories during the whole follow up. Slowness was more prevalent among those who were in the underweight or morbid obesity category. Exhaustion was more prevalent among those with morbid obesity consistently during the whole follow up and among those who were in the underweight category in the first 10 years of follow up. Low physical activity was more prevalent among those with morbidity obesity.

## Discussion

This study examined the relationship between BMI and frailty among non-frail older Mexican Americans at baseline who were followed over 18 years. This study showed a U-shaped relationship between BMI and frailty. Participants in the underweight category or obesity category II/morbid obesity were at 2 and 1.6 times, respectively, greater risk of frailty over time than those in the normal weight category, after controlling for all covariates. Results were similar after excluding pre-frail and frail participants at baseline. When we examined the relationship between BMI category and each frailty criterion, we found that underweight participants were at higher risk of weight loss, while the overweight or obese were at lower risk of weight loss. Those in the overweight or obesity categories were at higher risk for weakness, slowness, exhaustion, and low physical activity.

Our findings are similar to those from previous cross-sectional [6, 11–16] and longitudinal [18–25] studies that demonstrated a U-or J-shaped relationship between BMI and frailty. Gajic-Veljanoski et al. [41], using the Canadian Multi-Centre Osteoporosis Study, found that baseline BMIs $\geq$25 kg/m$^2$ was associated with faster frailty progression over 5-years with the greatest effect among those with BMI $\geq$40 kg/m$^2$ when compared to those with normal weight. Landré et al. [21] examined the relationship between weight history during adulthood and frailty among participants from the GAZEL (GAZ and ELectricité) Cohort and found that long term obesity and onset of obesity in late adulthood were associated with frailty over 25 years of follow up when compared with those with normal weight. In another study, Strandberg et al. [25] used the Helsinki Businessmen Study to determine whether midlife obesity could be a predictor of frailty over a 26-year follow up among the initially health. They found that those who were overweight or obese in midlife were twice as likely to develop frailty compared to those of normal weight. The studies of Landré [21] and Strandberg [25] considered normal weight as BMIs <25 kg/m$^2$. A recent systematic review and meta-analysis conducted by Yuan et al. [42] found that underweight and obesity both increased the risk of frailty over time.

Some mechanisms can explain our findings. The increased risk of frailty over time among those in the underweight category may be related to the weight loss criterion for frailty used in our study; however, weight loss can be seen in any of the BMI categories, not just in those who are underweight. Another explanation is the loss of muscle mass seen in older adults with undernutrition which is accompanied by low BMI [43, 44]. The increased risk of frailty over time among those in the obesity category II/morbid obesity may be related to the association of obesity with multiple conditions like insulin resistance, diabetes, cardiovascular disease, and increased inflammation, all of which are risk factors for frailty [45, 46]. Another explanation is related to the decreased muscle mass seen in older adults with obesity, known as "sarcopenia obesity"[47, 48].

Our study has some limitations. First, comorbid conditions were assessed through self-reports. This may lead to recall bias as compared to physician assessment. Second, participants excluded from the study were less healthy than those included, which might have led to underestimating the relationship between BMI and frailty. Third, participants who died before wave

2 may have produced a survival bias. Fourth, we do not have the measure of waist circumference (WC) assessed in all study waves. In older adults, WC has been suggested to be a better predictor than weight alone of whole-body fat percent and visceral adipose tissue [49, 50]. Fifth, our measure of frailty phenotype does not consider cognitive function or psychosocial measures [51]. Finally, our findings are not generalizable to the larger Hispanic population in the United States. This study has several strengths, including its large community-based sample of older Mexican Americans, a disadvantaged and underserved population, the length of follow up, and the use of generalized estimating equation models, an analytical approach that allows using all available data on socio-demographics, comorbidities, and BMI as time varying.

## Conclusions

This study shows that older Mexican Americans in the underweight or obesity type II/morbid obesity categories are at increased risk of frailty over an 18-year follow up when compared to those of normal weight. Overweight or obese older Mexican Americans were at higher risk of weakness, slowness, exhaustion, and low physical activity. Interventions should be implemented to improve body weight among the underweight and morbid obesity to enhance physical function, increase muscle strength, and increase levels of physical activity to prevent pre-frailty or frailty in this population.

## Author Contributions

**Conceptualization:** Megan Rutherford, Brian Downer, Chih-Ying Li, Lin-Na Chou, Soham Al Snih.

**Data curation:** Lin-Na Chou, Soham Al Snih.

**Formal analysis:** Lin-Na Chou, Soham Al Snih.

**Methodology:** Megan Rutherford, Brian Downer, Chih-Ying Li, Lin-Na Chou, Soham Al Snih.

**Project administration:** Soham Al Snih.

**Supervision:** Chih-Ying Li, Soham Al Snih.

**Validation:** Chih-Ying Li, Soham Al Snih.

**Visualization:** Megan Rutherford.

**Writing – original draft:** Megan Rutherford.

**Writing – review & editing:** Megan Rutherford, Brian Downer, Chih-Ying Li, Lin-Na Chou, Soham Al Snih.

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
