## [Decision Letter · Decision Letter 0]

20 Jun 2022

PONE-D-22-13598Body mass index and frailty among older Mexican Americans: Findings from an 18-year follow up.PLOS ONE

Dear Dr. Snih,

Thank you for submitting your manuscript to PLOS ONE. After careful consideration, we feel that it has merit but does not fully meet PLOS ONE’s publication criteria as it currently stands. Therefore, we invite you to submit a revised version of the manuscript that addresses the points raised during the review process.

We look forward to receiving your revised manuscript.

Kind regards,

Pasquale Abete

Academic Editor

PLOS ONE

Journal Requirements:

Additional Editor Comments:

According to decision of Reviewer #2 the manuscript needs a major revision.

Reviewers' comments:

Reviewer's Responses to Questions

**Comments to the Author**

1. Is the manuscript technically sound, and do the data support the conclusions?

Reviewer #1: Yes

Reviewer #2: Yes

2. Has the statistical analysis been performed appropriately and rigorously? 

Reviewer #1: Yes

Reviewer #2: Yes

3. Have the authors made all data underlying the findings in their manuscript fully available?

Reviewer #1: Yes

Reviewer #2: Yes

4. Is the manuscript presented in an intelligible fashion and written in standard English?

Reviewer #1: Yes

Reviewer #2: Yes

5. Review Comments to the Author

Reviewer #1: The Study evaluate the relationship between BMI and frailty among non-frail older Mexican Americans in a Longitudinal population-based study of 1,648 non-institutionalized Mexican Americans aged ≥ 67 years from the Hispanic Established Population for the Epidemiologic Study of the Elderly (1995/96-2012/13). Frailty phenotype was defined as meeting three or more of the following: unintentional weight loss of >10 pounds, weakness, self-reported exhaustion, low physical activity, and slow walking speed.

Participants with underweight or obesity category II/ morbid obesity had greater odds of frailty over time compared to those with normal weight after controlling for all covariates. Participants with BMIs in the overweight or category I obesity were at lower odds of frailty over time.

I found the study of interest conducted on a large cohort data. Data support conclusion. Figures and table are exhaustive. The statistical analysis is correct.

Reviewer #2: The authors examined the relationship between BMI and frailty among non-frail older Mexican Americans at baseline over 18 years of follow up in longitudinal population-based study of 1,648 non-institutionalized Mexican Americans aged ≥ 67 years from the Hispanic Established Population for the Epidemiologic Study of the Elderly (1995/96-2012/13). Frailty phenotype was defined as meeting three or more of the following: unintentional weight loss of >10 pounds, weakness, self-reported exhaustion, low physical activity, and slow walking speed. BMI (kg/m 2) was classified as underweight (<18.5), normal weight (18.5-<25), overweight (25-< 30), obesity category I (30-<35), and obesity category II/morbid obesity (≥35). Covariates included socio-demographics, comorbidities, cognitive function, and depressive symptoms. Generalized Estimating Equation models were performed to estimate the odds ratio (OR) and 95% confidence interval (CI) of frailty as a function of BMI category. Participants with underweight or obesity category II/ morbid obesity had greater odds of frailty over time compared to those with normal weight (OR 2.39, 95% CI 1.29-4.44 and OR 1.62, 95% CI 1.07-2.44, respectively) after controlling for all covariates. Participants with BMIs in the overweight or category I obesity were at lower odds of frailty over time.

The manuscript is interesting but I have some concerns that should be addressed. Firstly, frailty is actually considered as a “multidimensional” condition. The authors consider only the “physical” domain of the frailty. Please see and discuss “Abete P et al. The Italian version of the "frailty index" based on deficits in health: a validation study. Aging Clin Exp Res. 2017;29:913-926”. I suggest to add in the title “physical” before frailty.

Secondly, you have to consider the presence of “sarcopenic obesity” in your patients. Please see and discuss “Remelli F et al. Prevalence of obesity and diabetes in older people with sarcopenia defined according to EWGSOP2 and FNHI criteria. Aging Clin Exp Res. 2022 Jan;34(1):113-120.

6. PLOS authors have the option to publish the peer review history of their article (what does this mean?). If published, this will include your full peer review and any attached files.

Reviewer #1: No

Reviewer #2: No

---

## [Author Response · Author response to Decision Letter 0]

13 Jul 2022

Response to Reviewers' comments

Reviewer's Responses to Questions

Comments to the Author

1. Is the manuscript technically sound, and do the data support the conclusions?

Reviewer #1: Yes

Reviewer #2: Yes

Thank you.

2. Has the statistical analysis been performed appropriately and rigorously? 

Reviewer #1: Yes

Reviewer #2: Yes

Thank you.

3. Have the authors made all data underlying the findings in their manuscript fully available?

Reviewer #1: Yes

Reviewer #2: Yes

Thank you.

4. Is the manuscript presented in an intelligible fashion and written in standard English?

Reviewer #1: Yes

Reviewer #2: Yes

Thank you.

5. Review Comments to the Author

Reviewer #1: The Study evaluate the relationship between BMI and frailty among non-frail older Mexican Americans in a Longitudinal population-based study of 1,648 non-institutionalized Mexican Americans aged ≥ 67 years from the Hispanic Established Population for the Epidemiologic Study of the Elderly (1995/96-2012/13). Frailty phenotype was defined as meeting three or more of the following: unintentional weight loss of >10 pounds, weakness, self-reported exhaustion, low physical activity, and slow walking speed.

Participants with underweight or obesity category II/ morbid obesity had greater odds of frailty over time compared to those with normal weight after controlling for all covariates. Participants with BMIs in the overweight or category I obesity were at lower odds of frailty over time.

I found the study of interest conducted on a large cohort data. Data support conclusion. Figures and table are exhaustive. The statistical analysis is correct.

Response: Thank you!

Reviewer #2: The authors examined the relationship between BMI and frailty among non-frail older Mexican Americans at baseline over 18 years of follow up in longitudinal population-based study of 1,648 non-institutionalized Mexican Americans aged ≥ 67 years from the Hispanic Established Population for the Epidemiologic Study of the Elderly (1995/96-2012/13). Frailty phenotype was defined as meeting three or more of the following: unintentional weight loss of >10 pounds, weakness, self-reported exhaustion, low physical activity, and slow walking speed. BMI (kg/m 2) was classified as underweight (<18.5), normal weight (18.5-<25), overweight (25-< 30), obesity category I (30-<35), and obesity category II/morbid obesity (≥35). Covariates included socio-demographics, comorbidities, cognitive function, and depressive symptoms. Generalized Estimating Equation models were performed to estimate the odds ratio (OR) and 95% confidence interval (CI) of frailty as a function of BMI category. Participants with underweight or obesity category II/ morbid obesity had greater odds of frailty over time compared to those with normal weight (OR 2.39, 95% CI 1.29-4.44 and OR 1.62, 95% CI 1.07-2.44, respectively) after controlling for all covariates. Participants with BMIs in the overweight or category I obesity were at lower odds of frailty over time.

Comment: The manuscript is interesting, but I have some concerns that should be addressed. Firstly, frailty is actually considered as a “multidimensional” condition. The authors consider only the “physical” domain of the frailty. Please see and discuss “Abete P et al. The Italian version of the "frailty index" based on deficits in health: a validation study. Aging Clin Exp Res. 2017;29:913-926”. I suggest to add in the title “physical” before frailty.

Response: We agree with the reviewer comment. We have now added in the discussion under the study limitation section the following “Fifth, our measure of frailty phenotype does not consider cognitive function or psychosocial measures (52)” on page 15, lines 297-298.

Comment: Secondly, you have to consider the presence of “sarcopenic obesity” in your patients. Please see and discuss “Remelli F et al. Prevalence of obesity and diabetes in older people with sarcopenia defined according to EWGSOP2 and FNHI criteria. Aging Clin Exp Res. 2022 Jan;34(1):113-120. 

Response: We have added the following sentence in the discussion section “Another explanation is related to the decreased muscle mass seen in older adults with obesity, known as “sarcopenia obesity”(48, 49)” on page 14, lines 287-289.

---

## [Decision Letter · Decision Letter 1]

16 Aug 2022

PONE-D-22-13598R1Body mass index and frailty among older Mexican Americans: Findings from an 18-year follow up.PLOS ONE

Dear Dr. SNIH,

Thank you for submitting your manuscript to PLOS ONE. After careful consideration, we feel that it has merit but does not fully meet PLOS ONE’s publication criteria as it currently stands. Therefore, we invite you to submit a revised version of the manuscript that addresses the points raised during the review process.

We look forward to receiving your revised manuscript.

Kind regards,

Pasquale Abete

Academic Editor

PLOS ONE

Additional Editor Comments (if provided):

According to Reviewer's comments the manuscript needs a major revision.

Reviewers' comments:

Reviewer's Responses to Questions

**Comments to the Author**

1. If the authors have adequately addressed your comments raised in a previous round of review and you feel that this manuscript is now acceptable for publication, you may indicate that here to bypass the “Comments to the Author” section, enter your conflict of interest statement in the “Confidential to Editor” section, and submit your "Accept" recommendation.

Reviewer #2: (No Response)

2. Is the manuscript technically sound, and do the data support the conclusions?

Reviewer #2: No

3. Has the statistical analysis been performed appropriately and rigorously? 

Reviewer #2: Yes

4. Have the authors made all data underlying the findings in their manuscript fully available?

Reviewer #2: Yes

5. Is the manuscript presented in an intelligible fashion and written in standard English?

Reviewer #2: Yes

6. Review Comments to the Author

Reviewer #2: I did not found in the revised manuscript:

1) the modification of title: "I suggest to add in the title “physical” before frailty"

2) the references:

- #49: Remelli F et al. Prevalence of obesity and diabetes in older people with sarcopenia defined according to EWGSOP2 and FNHI criteria. Aging Clin Exp Res. 2022 Jan;34(1):113-120.

- #50: Abete P et al. The Italian version of the "frailty index" based on deficits in health: a

validation study. Aging Clin Exp Res. 2017;29:913-926”

7. PLOS authors have the option to publish the peer review history of their article (what does this mean?). If published, this will include your full peer review and any attached files.

Reviewer #2: No

---

## [Author Response · Author response to Decision Letter 1]

24 Aug 2022

Response to Reviewers' comments

Reviewer #2: 

Comment # 1: The modification of title: "I suggest to add in the title “physical” before frailty".

Response: The title now reads “Body mass index and physical frailty among older Mexican Americans: Findings from an 18-year follow up”

Comment # 2: The references:

- #49: Remelli F et al. Prevalence of obesity and diabetes in older people with sarcopenia defined according to EWGSOP2 and FNHI criteria. Aging Clin Exp Res. 2022 Jan;34(1):113-120.

- #50: Abete P et al. The Italian version of the "frailty index" based on deficits in health: a validation study. Aging Clin Exp Res. 2017;29:913-926”

Response: We have added the references. Reference number 48 correspond to “Remelli F et al. Prevalence of obesity and diabetes in older people with sarcopenia defined according to EWGSOP2 and FNHI criteria. Aging Clin Exp Res. 2022 Jan;34(1):113-120; and reference number 51 correspond to “Abete P et al. The Italian version of the "frailty index" based on deficits in health: a validation study. Aging Clin Exp Res. 2017;29:913-926”.

Thank you for noticing the absence of the references.

---

## [Decision Letter · Decision Letter 2]

26 Aug 2022

Body mass index and physical frailty among older Mexican Americans: Findings from an 18-year follow up.

PONE-D-22-13598R2

Dear Dr. SNIH,

We’re pleased to inform you that your manuscript has been judged scientifically suitable for publication and will be formally accepted for publication once it meets all outstanding technical requirements.

Kind regards,

Pasquale Abete

Academic Editor

PLOS ONE

Additional Editor Comments (optional):

No further comments.

Reviewers' comments:

Reviewer's Responses to Questions

**Comments to the Author**

1. If the authors have adequately addressed your comments raised in a previous round of review and you feel that this manuscript is now acceptable for publication, you may indicate that here to bypass the “Comments to the Author” section, enter your conflict of interest statement in the “Confidential to Editor” section, and submit your "Accept" recommendation.

Reviewer #2: All comments have been addressed

2. Is the manuscript technically sound, and do the data support the conclusions?

Reviewer #2: Yes

3. Has the statistical analysis been performed appropriately and rigorously? 

Reviewer #2: Yes

4. Have the authors made all data underlying the findings in their manuscript fully available?

Reviewer #2: Yes

5. Is the manuscript presented in an intelligible fashion and written in standard English?

Reviewer #2: Yes

6. Review Comments to the Author

Reviewer #2: The manuscript is really improves and all questions arised have been aswered. The manuscript is now acceptable to be published in PONE.

7. PLOS authors have the option to publish the peer review history of their article (what does this mean?). If published, this will include your full peer review and any attached files.

Reviewer #2: No

---

## [Editor Report · Acceptance letter]

31 Aug 2022

PONE-D-22-13598R2 

Body mass index and physical frailty among older Mexican Americans: Findings from an 18-year follow up. 

Dear Dr. Al Snih:

I'm pleased to inform you that your manuscript has been deemed suitable for publication in PLOS ONE. Congratulations! Your manuscript is now with our production department. 

Kind regards, 

on behalf of

Prof. Pasquale Abete 

Academic Editor

PLOS ONE